# SFHG-YOLO: A Simple Real-Time Small-Object-Detection Method for Estimating Pineapple Yield from Unmanned Aerial Vehicles

**DOI:** 10.3390/s23229242

**Published:** 2023-11-17

**Authors:** Guoyan Yu, Tao Wang, Guoquan Guo, Haochun Liu

**Affiliations:** 1School of Mechanical Engineering, Guangdong Ocean University, Zhanjiang 524088, China; yugy@gdou.edu.cn (G.Y.); wh214224@gmail.com (T.W.); guo2729359756@gmail.com (G.G.); 2Guangdong Provincial Engineering Technology Research Center for Marine Equipment and Manufacturing, Zhanjiang 524088, China; 3Southern Laboratory of Marine Science and Engineering (Guangdong Province), Zhanjiang 524013, China

**Keywords:** unmanned aerial vehicle, small object detection, lightweight network, adaptive contextual information fusion, high-density object detection, deep learning

## Abstract

The counting of pineapple buds relies on target recognition in estimating pineapple yield using unmanned aerial vehicle (UAV) photography. This research proposes the SFHG-YOLO method, with YOLOv5s as the baseline, to address the practical needs of identifying small objects (pineapple buds) in UAV vision and the drawbacks of existing algorithms in terms of real-time performance and accuracy. Field pineapple buds are small objects that may be detected in high density using a lightweight network model. This model enhances spatial attention and adaptive context information fusion to increase detection accuracy and resilience. To construct the lightweight network model, the first step involves utilizing the coordinate attention module and MobileNetV3. Additionally, to fully leverage feature information across various levels and enhance perception skills for tiny objects, we developed both an enhanced spatial attention module and an adaptive context information fusion module. Experiments were conducted to validate the suggested algorithm’s performance in detecting small objects. The SFHG-YOLO model exhibited significant gains in assessment measures, achieving mAP@0.5 and mAP@0.5:0.95 improvements of 7.4% and 31%, respectively, when compared to the baseline model YOLOv5s. Considering the model size and computational cost, the findings underscore the superior performance of the suggested technique in detecting high-density small items. This program offers a reliable detection approach for estimating pineapple yield by accurately identifying minute items.

## 1. Introduction

The third-most-eaten fruit in the world, the pineapple, is a well-known tropical fruit that is adored by customers everywhere [1]. It is a popular option on the world fruit market because of its distinctive flavor and abundance of volatile chemicals. Accurate yield estimation is crucial for those involved in pineapple growing. Forecasting and predicting pineapple yields are necessary for effective agricultural planning and decision-making to appropriately plan cultivation areas and fertilizer and pesticide usage, boosting productivity and agricultural efficiency [2,3]. Furthermore, accurate yield assessment helps producers satisfy market demands, take advantage of sales possibilities, and improve income prospects by contributing to market projections and supply–demand balance.

With the rapid development of drone technology, specialized or general-purpose drones equipped with cameras have been widely applied in various fields such as agriculture, aerial photography, public safety, and ecological conservation [4]. The aim of this study was to leverage drone technology in developing a pineapple-detection system to meet the precise estimation needs of pineapple yield. Figure 1 illustrates a concise workflow of the pineapple-yield-estimation system we have constructed. The system acquires relevant data through drone flights. Accurate statistical assessment of pineapple yield typically requires estimation during the pineapple’s flowering period to achieve precision and effectiveness. Therefore, it is essential to conduct flights at the appropriate growth stage. By constructing a pineapple-flowering-stage target-detection network, the system recognizes pineapples at this stage and employs a counting algorithm to achieve real-time quantification of the identified pineapple targets. The density information of pineapple plants is obtained by counting each detected pineapple plant. Finally, the acquired pineapple plant density data are input into the yield estimation model to achieve pineapple yield estimation. The yield estimation model is based on statistical methods or other machine learning algorithms, integrating pineapple density and other agricultural features to calculate the yield. The main focus of this research is on enhancing existing research methods while fully considering the unique applications of drones, to develop a real-time and highly accurate small-target-detection-algorithm.

In response to the challenges of pineapple yield estimation and the rapid development of deep learning technology, convolutional-neural-network (CNN)-based object-detection algorithms have been employed to detect objects in drone images. Therefore, this study adopts a deep-learning-based object-detection approach to enhance both detection accuracy and real-time performance, thereby establishing the foundation for subsequent pineapple yield estimation. Currently, within the realm of deep-learning-based object-detection algorithms, two primary categories exist: two-stage and one-stage object-detection algorithms. Two-stage object detection algorithms [5,6,7,8] employ two distinct steps for object detection. Firstly, they utilize a region proposal network (RPN) like the one in Faster R-CNN to extract potential object regions. Subsequently, these regions are fed into networks for object classification and bounding box regression to obtain the final detection results. Such algorithms generally exhibit high accuracy due to their ability to refine the detection process gradually through two independent stages, enhancing the precision of object localization. However, as these processes involve two stages, their computational speed is relatively slow, making them less suitable for scenarios with stringent real-time requirements. In contrast, one-stage object-detection algorithms [9,10,11,12,13] directly complete all steps of object detection, including object classification and bounding box regression, within a single neural network. This characteristic renders one-stage algorithms more efficient in terms of processing speed, making them suitable for applications demanding real-time performance. Nevertheless, because of their direct prediction of object properties, they may exhibit relatively coarser object localization compared to two-stage methods, leading to some degree of localization error. Despite having lower predictive accuracy than two-stage object-detection algorithms, the You Only Look Once (YOLO) algorithm, known for its comprehensive performance, can be utilized for detecting objects in drone images.

However, there are several concerns to take care of while conducting target detection with drones, including tiny object detection, complicated backdrop interference, perspective and scale fluctuations, occlusion issues, and the need for real-time processing. Researchers have undertaken several experiments on target recognition in UAV-captured situations as a solution to these problems. The spatial attention module (SAM) and channel attention module (CAM) were combined to create the spatial channel attention model (SCAM) by Shuai et al. [14]. They changed the link between SAM and CAM, compressed features to improve the fully connected layer in CAM, and applied SCAM to YOLOv5 to increase the spatial dimension feature capture. This not only lightened the strain on the computer, but it also somewhat increased accuracy. To improve the capacity to recognize heavily occluded objects in UAV pictures, Zhu et al. [15] developed TPH-YOLOv5, an upgrade of YOLOv5 that included a prediction head and integrated transformer encoder blocks (TPH) into the head component. To suppress inconsistent gradient computations in the feature pyramid network (FPN), Hong et al. [16] proposed the scale selection pyramid network (SSPNet), which uses the context attention module (CAM), scale enhancement module (SEM), and scale selection module (SSM) to control data flow between adjacent layers. Zhao et al. [17] improved the network’s generalization ability through data cleaning and augmentation and introduced prior anchor frames to reconstruct the detection layer’s confidence loss function based on the intersection over union (IoU). This was performed to address the issue of false positives and false negatives caused by occlusion. To solve the issue of the network not effectively concentrating on tiny objects, Zhang et al. [18] introduced a new detection network called DCLANet for cropping and local processing of dense small objects in UAV pictures. By developing a technique for tiny object recognition in UAV views, improving Resblocks and the overall darknet structure, and training using optimized datasets gathered from UAV viewpoints, Liu Mingjie et al. [19] enhanced the YOLOv3 algorithm. To enhance the precision of tiny item recognition, other researchers have also tried a variety of methods, including multi-scale training [20], multi-branch parallel feature pyramid networks [21], cascaded architectures [22], and data augmentation [23]. Although these techniques have somewhat improved tiny item identification, real-time performance is still a problem in embedded systems with limited resources.

The goal of this study was to design a lightweight pineapple-red-dot target-detection network based on YOLOv5s as the baseline to further improve the accuracy of target detection in UAV-captured scenes and address the issues of poor detection performance caused by small-sized and densely packed targets, as well as the requirement for real-time processing. The network attempts to increase target detection’s precision and real-time performance in scenes filmed by UAVs. An adaptive context information fusion and spatial attention improvement module was created to improve the network’s perception of contextual information and its focus on tiny targets to do this. The network’s detection accuracy and real-time performance are simultaneously improved by enhancing the backbone network, using more-successful data-augmentation procedures, and introducing a specific head structure for tiny target identification. This work provides shallower feature maps as a separate layer for tiny target identification to improve the detection performance of targets at medium and long ranges. In conclusion, the study’s main contributions are as follows:An adaptive multi-scale context information fusion module was created, allowing the network to independently choose relevant feature information helpful for tiny object recognition and improving the precision of small object detection.A more-advanced spatial attention module was created to increase the network’s capacity to perceive information about small items, sharpen the network’s focus on them, and ultimately increase the accuracy of small object recognition.This research obtained positive outcomes by outperforming the baseline detection network in terms of tiny-item-detection performance using a number of enhancement approaches.

The subsequent sections of this study are divided into the following four parts. In Section 2, we provide a concise overview of the YOLOv5 algorithm and propose several improvements based on it, introducing two novel modules. Section 3 focuses on the process of data collection, the selection of evaluation metrics, and a thorough analysis of the experimental results. Finally, Section 4 delves into future research directions and offers a comprehensive analysis of the findings of this study.

## 2. Proposed Method

### 2.1. Introduction to YOLOv5 Algorithm

The one-stage object-identification system called YOLOv5 has significantly outperformed YOLOv4 in terms of speed and accuracy [12]. Its improvements mainly concentrate on the following four areas: The mosaic-data-augmentation operation, which improves model training speed and network accuracy, is the first new feature introduced by YOLOv5. The model may acquire more contextual information and simultaneously train on numerous pictures by integrating several photos into a bigger mosaic image, which increases training efficiency. Second, to efficiently handle targets of various sizes, YOLOv5 uses adaptive anchor computation and adaptive picture scaling techniques. This method improves the model’s capacity to generalize and adapt by better coping with targets of different scales. Thirdly, YOLOv5 enhances the backbone network by drawing inspiration from fresh concepts in other detection techniques. To simplify the computation and improve feature representation, it adds the focus structure and the cross-stage partial (CSP) structure. The Neck network uses an FPN+PAN structure and adds the CSP2 structure, which was inspired by the CSPnet architecture. This improves object identification performance and strengthens feature fusion capabilities. The effectiveness of YOLOv5 in detecting objects is greatly improved by these actions. While YOLOv5 is excellent at recognizing large items, finding tiny things is still challenging due to issues with object localization, identification uncertainty, and a lack of scale information. Therefore, it is not possible to match our unique criteria by simply applying YOLOv5 to small-sized photos taken by Jetson Xavier NX embedded devices installed on UAVs. Further enhancements and adjustments are required for YOLOv5 to solve the difficulties of recognizing tiny objects and satisfy our particular requirements.

### 2.2. Improved YOLOv5

To strike a balance between real-time requirements and computational complexity, YOLOv5s was chosen as the baseline model in this study. YOLOv5s, a lightweight model, to some extent, satisfies real-time demands and has been thoroughly optimized, boasting high inference speed. However, considering the characteristics of UAV-captured images and the task requirements, directly introducing custom modules to YOLOv5s would escalate computational intricacy, thereby undermining real-time performance. Concurrently, the additional augmentation of model storage space and computational load would also exert adverse effects on real-time capability. Consequently, to uphold real-time performance while enhancing detection accuracy, this research enhanced the baseline model in the following key aspects: (1) employing judicious data augmentation techniques to bolster the model’s generalization capability, enabling it to adapt to various scenes and changing conditions; (2) reconfiguring the YOLOv5s backbone network by incorporating the MobileNetv3 [24] Bneck module to expedite inference speed and curtail computational complexity; (3) introducing an adaptive context information fusion module adept at assimilating context information from diverse scales to heighten detection precision; (4) proposing a spatial attention module to amplify the model’s focus on spatial information, thereby refining the detection accuracy of small objects; (5) augmenting the detection precision for small objects, appending a dedicated small object detection head; (6) implementing the VariFocal loss [25] function to address data imbalance and bounding box regression accuracy issues; (7) introducing the efficient IoU loss [26] (EIoU) function to enhance bounding box regression accuracy. The optimized model is depicted in Figure 2.

#### 2.2.1. Reasonable Data Augmentation

The model’s resilience, generalization capacity, and training performance may all be enhanced with reasonable data augmentation, making it more capable of adjusting to different input variations and scene needs. Excessive data augmentation, on the other hand, might make training more computationally difficult and take longer, while having a smaller effect on performance gain. To balance the impacts of data augmentation and computational burden, we concentrated on the illumination intensity and the UAV’s tilt angle in this investigation. We used a variety of random data-augmentation techniques, such as random scaling, random brightness modification (contrast and saturation), random cropping, random flipping, and random rotation, to improve the model’s resilience and generalizability and lower the danger of overfitting. These methods enhance the model’s ability to adapt to various sizes, orientations, and rotations, as well as diversify its skills in recognizing small objects. We modified the image scale using random scaling so that the model can recognize objects of various sizes. The model is better able to respond to changes in illumination since random brightness modification replicates various lighting circumstances. By introducing local information about the objects, random cropping helps the model learn specific properties. Random rotation replicates the UAV’s attitude changes, boosting the model’s detection of rotated objects. Random flipping enhances data diversity, improving the model’s capacity to recognize objects in horizontal and vertical orientations. By using these realistic data augmentation techniques, we expanded the model’s training samples, strengthening the model’s capacity to adapt to various scenarios and settings and, so, improving its resilience and generalization.

#### 2.2.2. Lightweight Backbone

The weight of the model can be reduced in actual projects by substituting the backbone network. CSPDarkNet 53 was the initial backbone network for YOLOv5. In this study, MobileNetv3 was chosen as the YOLOv5 backbone network. In order to obtain the best configuration from a collection of discrete options, MobileNetv3 uses reinforcement learning and MnasNet [27] to conduct a coarse-grained architectural search. We then used NetAdapt [28] to fine-tune this setup, preserving a little performance hit while adjusting unused activation channels. The squeeze-and-excitation (SE) [29] attention mechanism is a new addition to MobileNetv3’s design. However, the SE module ignores the significance of positional information and only takes into account information encoding across channels. Positional information is essential for the performance of the model in many visual tasks that call for collecting the structural information of objects. We present the coordinate attention module (CA) [30] to overcome this problem. The CA module, in contrast to the SE module, uses a more-effective method to record positional data and connections between channels in the feature map, improving the network’s capacity to represent features.

The CA module leverages precise positional information to encode channel relationships and long-term dependencies (Figure 3). This module consists of two crucial steps: coordinate information embedding and coordinate attention generation (illustrated in Figure 4). When employing global pooling techniques, preserving positional information becomes challenging. To address this issue, we utilized the following approach: Firstly, we decomposed global pooling into a series of 1D feature encoding operations (as shown in Equations (2) and (3)) for a given input, *X*. Specifically, we applied pooling kernels of dimensions (*H*, 1) and (1, *W*) separately along the vertical and horizontal directions to encode channels. Equation (2) denotes the output for channel *c* with a height of *h*, while Equation (3) represents the output for channel *c* with a width of *w*. These two transformations aggregate features along distinct directions, yielding a pair of direction-aware feature maps. Simultaneously, the attention module captures long-range dependencies along one spatial direction while retaining precise positional information along the other spatial direction. This aids the network in accurately localizing regions of interest.
(1)$$Z_{c}=\frac{1}{H\times W}\sum_{i=1}^{H}\sum_{j=1}^{W}X_{c}\left(i,j\right)$$
(2)$$Z_{c}^{h}\left(h\right)=\frac{1}{W}\sum_{0\leq i<W}X_{c}\left(h,i\right)$$
(3)$$Z_{c}^{w}\left(w\right)=\frac{1}{H}\sum_{0\leq i<H}X_{c}\left(j,w\right)$$

Furthermore, to achieve both global awareness and precise positional information, the two components are cascaded and then transformed using a convolutional function. This integration maximizes the utilization of positional information, enabling the model to accurately capture target regions while effectively capturing inter-channel relationships. Equation (4) represents the final output of the CA module. The enhanced BneckCA module is depicted in Figure 4.
(4)$$Z_{out}=X_{c}\left(i,j\right)⊗G_{c}^{h}\left(i\right)⊗G_{C}^{w}\left(j\right)$$

The network topology of MobileNetv3 before the 32 × downsampling was kept to replace the YOLOv5 backbone network and satisfy the structural design specifications of the object-detection model, as shown in the shaded region of Figure 2. With minimal computational complexity, the replacement network can more easily adapt to the object-detection job. The network is more suited for our object-identification job as a result of this change, which takes into consideration both model performance and computational economy. Keeping the network structure before 32-× downsampling allows for the use of MobileNetv3 features while keeping a lower computational cost, better meeting the needs of the object-detection job.

#### 2.2.3. Adaptive Context Information Fusion Module

Effectively leveraging contextual information is of paramount importance in the context of small object detection. Small objects typically exhibit limited pixel-level details and subtle characteristics, and relying solely on local features might pose challenges. However, by capitalizing on contextual cues across multiple scales, a more-comprehensive background can be supplied. This augmentation enhances the capacity to comprehend and discriminate small objects, thereby elevating detection performance in aspects such as target localization, object discrimination, scale adjustment, and contextual relationships.

Low-level features are particularly significant in tiny item recognition because they include edge and texture data, which can capture object borders and specifics. On the other hand, high-level characteristics reflect item categories and general forms and include more-sophisticated semantic information. They can record a wider variety of contextual data, such as the overall organization and contextual connections between individual items. The requirements of tiny object detection, however, cannot be satisfied by relying simply on the fusion of low-level information (as in YOLOv5s). As a result, efficient feature information fusion at both the low and high levels is required. As illustrated in Figure 5, our study developed an adaptive context information fusion module (ACIFM) to overcome this problem. The goal of this module is to combine characteristics from several levels to improve the model’s comprehension of the overall information and picture structures, eventually enhancing accuracy and resilience in the detection job.

Figure 5 illustrates the process of information fusion from three distinct hierarchical scales. Given the input $\chi =\chi_{1},\chi_{2},\chi_{3}$, initially, we employed three separate 1 × 1 convolutions to compress the channels, resulting in the compressed features $\chi_{1}^{\prime },\chi_{2}^{\prime },\chi_{3}^{\prime }$, aimed at reducing computational load. Subsequently, through a series of upsampling operations, we progressively integrated high-level features upwards, yielding preliminary fused feature information $\gamma_{1},\gamma_{2},\gamma_{3}$. These fused features amalgamate insights from both low-level and high-level strata, aiding the network in more precisely localizing small objects. In particular, the high-level information encapsulates advanced semantics, assisting the network in distinguishing and delineating disparities between small objects and their surrounding environment, thereby augmenting the discernment capability of small objects. The computational expressions for $\gamma_{1},\gamma_{2},\gamma_{3}$ are detailed in Equations (5)–(7). Here, “Conv1” signifies a 1 × 1 convolution, employed to further extract effective information and adjust the channel dimensions, while “Up” denotes the upsampling operation.
(5)$$\gamma_{1}=Conv1\left(Up\left(\gamma_{3}\right)\right)+Conv1\left(Up\left(\gamma_{2}\right)\right)+\chi_{1}$$
(6)$$\gamma_{2}=Conv1\left(Up\left(\gamma_{3}\right)\right)+\chi_{2}$$
(7)$$\gamma_{3}=\chi_{3}$$

Given the significance of integrating information across different scales post-fusion, we introduced learnable layers to enable the network to autonomously learn the weight coefficients between these distinct scales. The specific computations are outlined in Equations (8) and (9). Through this approach, the model can independently determine the importance of various scale-related information based on task requirements and data characteristics and adaptively adjust the weight coefficients. Consequently, this allows for more-effective utilization of diverse-scale information and facilitates the efficient fusion of critical insights.
(8)$$\nu =Concat{\left(GAP\left(\gamma_{1}\right),GAP\left(\gamma_{2}\right),GAP\left(\gamma_{3}\right)\right)}_{dim=1}$$
(9)$$\left[W_{1},W_{2},W_{3}\right]=\Gamma \left(\nu \right)$$

In the equations, *GAP* stands for the global average pooling operation, where *dim* = 1 signifies concatenating three tensors derived by global average pooling along the channel dimension. The learnable weight layer is shown as Γ. Equations (10)–(12) show how the weight coefficients $W_{1},W_{2},W_{3}$, which stand for the importance of various scales, are multiplied elementwise with the relevant tensors $\gamma_{1},\gamma_{2},\gamma_{3}$ before being added to obtain the result. Equation (13) displays the module’s ultimate result, where Conv3 denotes a 3 × 3 convolution. The technique enhances the accuracy and resilience of tiny item recognition without considerably increasing processing resources by integrating data from higher to lower hierarchical levels. This makes it possible for the model to handle object identification tasks at various sizes with greater adaptability and flexibility.
(10)$$\beta_{1}=\gamma_{1}⊗W_{3}$$
(11)$$\beta_{2}=Up\left(Conv1\left(\gamma_{2}\right)\right)⊗W_{2}$$
(12)$$\beta_{3}=Up\left(Conv1\left(\gamma_{3}\right)\right)⊗W_{1}$$
(13)$$out=Conv3(\beta_{1}+\beta_{2}+\beta_{3})$$

#### 2.2.4. Improved Spatial Attention Module

Spatial attention modules are widely employed in computer vision tasks, primarily designed to guide neural networks in allocating varying degrees of attention and processing to different regions of the input images. This serves to enhance the model’s focus on areas of interest, facilitating a more-effective capture of crucial spatial features. In the context of small object detection, spatial attention modules play a pivotal role. Small objects typically possess reduced spatial dimensions and pixel intensities, making them prone to being overshadowed by the image background. Spatial attention modules can augment the model’s perceptual capabilities and detection accuracy for small objects. By assigning weights to distinct locations, this module amplifies the emphasis on target regions, consequently bolstering the network’s detection prowess.

In this study, an enhanced spatial attention structure was devised (as depicted in Figure 6). Diverging from the spatial attention component of the convolutional block attention module (CBAM) [31] mechanism, we introduced an alteration after global average pooling and global max pooling. Specifically, we incorporated 1×1 convolution post-pooling operations and amalgamated their outputs to attain a more-enriched feature representation. This approach sought to amplify the model’s comprehension of the input feature map and aid the attention module in better distinguishing and focusing on diverse feature information. Furthermore, via a 7×7 convolution operation, spatially correlated information was extracted from the input feature map using horizontal and vertical directions. This information was then multiplied with the input feature map, enhancing the representational capacity of the region of interest. To be precise, the information from the horizontal and vertical dimensions captured spatial relationships between pixels within the input feature map. Employing convolution operations on these dimensions enabled a more-accurate capture of the spatial patterns and structures within the feature map. This step facilitated the differentiation of features in distinct locations. For instance, in object detection tasks, various object positions might entail varying contextual information and significance. Leveraging information from the horizontal and vertical dimensions allowed for a more-precise allocation of attention and the reinforcement of pivotal feature representations. By capitalizing on these-dimensional cues, a more-refined focus on the intricate characteristics of small objects was achieved, thereby elevating detection accuracy and robustness.

Figure 6 illustrates the enhanced spatial attention module. Assuming an input tensor X (with dimensions of C×H×W), the input first undergoes global maximum pooling, global average pooling, and 1×1 convolution operations, yielding three tensors of size 1×H×W each. These tensors are concatenated along the channel dimension to obtain a tensor of size 3×H×W. Subsequently, this tensor undergoes a convolution operation with a kernel size of 7×7, followed by additional convolutions along the vertical and horizontal directions, with kernel sizes of 1×H and W×1, respectively. By fusing information from both horizontal and vertical dimensions, a more-comprehensive spatial context is provided, assisting the spatial attention module in better understanding and capturing the spatial relationships within the input feature map. Finally, the information from the horizontal and vertical dimensions is multiplied to yield a tensor of size 1×H×W. This tensor is then normalized using the Sigmoid function to obtain the final spatial attention feature map. Such feature maps contribute to enhancing the network’s focus on the target regions, thereby improving its ability to process regions of interest. The calculation process can be referred to as per Equations (14)–(16), where maxpool denotes the maximum pooling operation, avgpool denotes the average pooling operation, Conv represents the convolution operation, Dwise denotes the depthwise separable convolution operation, and σ represents the Sigmoid function.
(14)$$Z_{3\times H\times W}=Concat{\left(maxpool\left(X\right),avgpool\left(X\right),Conv{\left(X\right)}_{1\times 1}\right)}_{dim=1}$$
(15)$$Z_{1\times H\times W}=\sigma \left(Conv{\left(Dw{\left(Z_{3\times H\times W}\right)}_{7\times 7}\right)}_{1\times H}⊗Conv{\left(Dw{\left(Z_{3\times H\times W}\right)}_{7\times 7}\right)}_{W\times 1}\right)$$
(16)$$Z_{O}=X⊗Z_{1\times H\times W}$$

#### 2.2.5. Addition Detection Head

Taking into consideration the size of small objects, we made targeted improvements to YOLOv5 by introducing a dedicated head module designed for small object detection. The aim was to enhance the model’s performance and robustness in detecting small objects. Specifically, the addition of a specialized small-object-detection head allows for better adaptation to the size range of small objects. By incorporating additional convolutional layers and adjusting parameters within the small object detection head, we can enhance the model’s perception and feature representation capabilities for small objects. Another purpose of adding the small object detection head is to enable more-refined adjustment and training specifically for small objects. Given the relatively small size and features of small objects, they can be challenging to differentiate and detect. Thus, the training process may require more-meticulous annotation and loss function design. Through a dedicated small-object-detection head, adjustments and optimizations can be made to address the unique requirements of small objects, thereby improving the detection accuracy and recall rate for these objects.

In deep learning networks, the loss function is essential because it quantifies the discrepancy between the model’s expected outcomes and the actual labels, generating gradient signals for parameter updates and optimization. We frequently run across the problem of unbalanced positive and negative samples while working with tiny object datasets, where the background sample count is significantly higher than the goal sample count. This imbalance may hinder the network’s ability to converge, forcing it to concentrate more on categorizing background samples while ignoring the learning and detection of tiny objects, which may negatively influence the performance of the model.

To address the issue of imbalanced positive and negative samples, we introduced the VariFocal loss function (Equation (17)) in our approach. The VariFocal loss incorporates a focusing parameter that dynamically adjusts the focus coefficients for different samples, assigning higher weights to challenging-to-detect small objects within the loss function. Small objects typically exhibit lower signal-to-noise ratios and fewer discernible features, rendering them relatively harder to detect. By adopting the focusing coefficients, the VariFocal loss increases the emphasis on detecting small objects, thereby enhancing their recall rate and detection accuracy. This loss function dynamically adjusts the weights for each sample, assigning greater weights to rare categories (such as the category to which small objects belong) within the loss function. This weight adjustment strategy balances the importance across categories, reinforcing the learning of the small object category and, consequently, improving the detection capabilities for such objects. Through its mechanism of focal adjustment and category balancing, the VariFocal loss enhances the model’s performance in small-object-detection tasks. By boosting the recall rate of small objects, reducing instances of missed detection, and lowering false positive rates, the VariFocal loss significantly enhances the accuracy of small object detection.
(17)$$VFL\left(p,q\right)=\left\{\begin{array}{cc} -q(q\log(p))+(1-q)\log(1-p), & q>0\\ -\alpha p^{\gamma }\log\left(1-p\right),q=0 & \end{array}\right.$$

#### 2.2.6. Bounding Box Regression

In object detection, a commonly used evaluation metric is the intersection over union (IoU), which measures the degree of overlap between predicted bounding boxes and ground truth bounding boxes. However, for small object detection, the traditional IoU has some drawbacks. Firstly, the IoU is not scale-sensitive, meaning that, even with a certain degree of positional error in the bounding boxes, the IoU calculation still yields a low result, making accurate localization of small objects challenging. Secondly, the calculation of the IoU is susceptible to the issue of sample imbalance, where the number of background samples far exceeds that of target samples. This bias towards background samples can lead the model to prioritize the classification of the background, thus neglecting the learning and detection of small objects. Additionally, the IoU is sensitive to positional errors in the bounding boxes, causing even slight deviations in position to lower the IoU calculation result.

To alleviate the limitations of the traditional IoU and enhance the accuracy of small object detection, we introduced an improved method for computing the intersection over union, namely the efficient IoU (EIoU) loss (as shown in Equation (18)). The EIoU takes into consideration more factors, including scale, positional deviation, and shape consistency, providing a more-comprehensive reflection of the overlap between the object and the ground truth bounding box. Compared to the traditional IoU, the EIoU loss is more sensitive to changes in scale and position. Even in the presence of positional errors, it can yield higher EIoU values, thereby enhancing the model’s ability to accurately localize small objects. Furthermore, the EIoU loss helps mitigate the issue of sample imbalance, directing the model’s attention toward learning and detecting small objects more effectively, thus enhancing its performance in scenarios involving small objects. Therefore, introducing the EIoU as an evaluation metric can improve the performance of small object detection and enhance the model’s perception and accuracy in detecting small objects.
(18)$$L_{EIoU}=1-IoU+\frac{\rho^{2}\left(b,b^{gt}\right)}{c^{2}}+\frac{\rho^{2}\left(w,w^{gt}\right)}{c_{w}^{2}}+\frac{\rho^{2}\left(h,h^{gt}\right)}{c_{h}^{2}}$$

## 3. Experimental Results and Analysis

### 3.1. Dataset

For data collection in this investigation, we used a Fenghang Corporation FH ZD680PRO drone. The drone flies according to the base station’s commands while filming, and taking pictures during its flights. We chose pineapple fruits during the blossoming stage as the main objects of attention for the research goals. The facility is located in Xuwen County, Zhanjiang City, Guangdong Province. The UAV shown in Figure 1 is one of the devices used in the tests.

### 3.2. Evaluation Metric

In this experiment, we used the accuracy, recall, mean Average Precision at an IoU threshold of 0.5 (mAP@0.5), and mean average precision at a IoU threshold of 0.5 to 0.95 (mAP@0.5:0.95) as the assessment criteria for model performance. The accuracy of a network model or classifier is measured by how many of the predicted positive samples are actually true positive samples out of all the positive predictions. Model performance is improved by a greater accuracy value. Recall is the percentage of authentically positive samples among all the model-predicted positive samples. Since accuracy and recall are two variables that are connected to one another, while precision is high, recall is typically poor, and vice versa.
(19)$$Precision=\frac{TP}{TP+FP}$$
(20)$$Recall=\frac{TP}{TP+FN}$$

Average precision (AP) calculates the precision values corresponding to various recall levels and then averages those precision values. Typically, AP values range from 0 to 1, where higher values indicate better performance. mAP is the average of AP values across multiple categories. mAP@0.5 represents the average precision for all categories when the IoU threshold is set to 0.5. mAP@0.5:0.95 refers to the average precision computed across different IoU thresholds (e.g., 0.5, 0.55, 0.6, …, 0.95). In this case, *C* stands for the number of classes, *N* for the quantity of IoU thresholds, and *k* for the actual IoU threshold. The terms accuracy and completeness (P and R, respectively) correspond to the model’s precision and recall.
(21)$$AP=\sum_{k=1}^{N}P\left(k\right)\Delta r\left(k\right)$$
(22)$$mAP=\frac{1}{C}\sum_{i=1}^{C}AP_{i}$$

Furthermore, we assessed the network’s performance from four additional aspects. These included the unit time detection count (frames per second (FPSs)), employed to evaluate the model’s detection speed, the floating-point operations per second (FLOPs), used to assess the model’s computational complexity, the model parameter count for evaluating the model size, and the model memory requirements (MMRs) to further appraise the model size. Finally, to accelerate inference, we employed CUDA to manage the entire detection pipeline.

### 3.3. Experiment Settings

The hardware and software tools utilized in the experimental environment are listed in Table 1. Table 2 displays the settings for key parameters used in the experiment’s training phase.

### 3.4. Comparisons with SOTAs

To evaluate the performance of the proposed algorithm in terms of detection accuracy and speed, with a focus on highlighting the enhancements brought by the improved SFHG-YOLO algorithm in the context of small object detection, we conducted a comprehensive comparative analysis. We compared the enhanced SFHG-YOLO algorithm against various other algorithms, including YOLOv3, YOLOv3-Tiny, YOLOv4, YOLOv4-Tiny, YOLOv5, YOLOv6 [32], YOLOv7 [33], YOLOv7-Tiny, YOLOv8, and YOLOX [34]. All algorithms were evaluated using their minimal network architectures. Through contrasting experimental outcomes across different algorithms, our objective was to unveil the significant improvements of the SFHG-YOLO algorithm for small object detection. We placed particular emphasis on the two key metrics of detection accuracy and speed to comprehensively assess the algorithm’s performance. Comparative results are presented in Table 3, Figure 7 and Figure 8. Furthermore, we illustrate the precision–recall (PR) curve (Figure 9) to conduct a more-detailed analysis of the algorithm’s performance.

Table 3 presents a comparison between our improved SFHG-YOLO model and other state-of-the-art algorithms. Firstly, we focused on the mAP metric as a crucial measure of object-detection performance. Through the analysis of the experimental results, the SFHG-YOLO model demonstrated outstanding performance on both the mAP@0.5 and mAP@0.5:0.95 metrics. Specifically, for the mAP@0.5 criterion, the SFHG-YOLO model achieved a high level of 0.864, significantly surpassing other models. This implies that the SFHG-YOLO model excels in accurately detecting target objects. Furthermore, within the stricter IoU threshold range of 0.5 to 0.95, the SHFP-YOLO model achieved a mAP@0.5:0.95 of 0.42, further highlighting its excellent object-detection precision. This indicates that the SFHG-YOLO model maintains a high accuracy even when handling higher IoU thresholds. In addition, we also examined the model’s inference speed and memory requirements, using the FPSs for speed evaluation and the MMRs to reflect each model’s memory consumption. The experimental results revealed that YOLOV3-tiny excelled in FPS performance, reaching 238 frames. In comparison, the SFHG-YOLO model achieved an FPSs of 98 frames. However, despite a speed reduction, the SFHG-YOLO model still maintained a higher object detection accuracy, particularly at higher IoU thresholds. Notably, the comparison highlights that SHFP-YOLO possesses the smallest MMR value among all models, indicating its prominence in terms of model size and memory consumption. Therefore, in scenarios demanding higher accuracy, the SFHG-YOLO model remains a noteworthy choice.

Additionally, we plotted curves illustrating the performance of different models at various thresholds for mAP@0.5 and mAP@0.5:0.95 to visually showcase their performance under different conditions. From Figure 7, it is evident that the SFHG-YOLO model demonstrates superior performance in terms of mAP@0.5 compared to the other models. Its curve exhibits a more-pronounced upward trend, indicating enhanced object detection capability at lower thresholds. On the other hand, in the case of mAP@0.5:0.95, the curve of the SHFP-YOLO model remains relatively flat, signifying its ability to maintain high accuracy even at higher IoU thresholds. Figure 9 presents a comparison of the PR curves for different algorithms, with recall plotted on the horizontal axis and precision on the vertical axis. The area enclosed by the PR curve and the axes represents the AP value. The PR curve of the SFHG-YOLO algorithm almost entirely encompasses the PR curves of the other algorithms, highlighting the superior performance of our proposed approach. Based on the insights gleaned from the current curves, it is evident that the SFHG-YOLO model consistently outperformed other models across various thresholds. In summary, as indicated by the experimental results, the SFHG-YOLO model exhibited commendable performance in the object-detection tasks. It excelled in the mapped metric, demonstrating superior average precision across different IoU thresholds compared to the other models evaluated in the experiments. Despite a slight reduction in inference speed, the SFHG-YOLO model strikes a well-balanced compromise between accuracy and speed, rendering it highly practical for numerous real-world applications.

#### VariFocal Loss Function

In order to verify the algorithm’s effectiveness, we also made visual comparisons of the results, and Figure 10 amply demonstrates the improved algorithm’s considerable improvements in object-detecting tasks. We hid the labels of the detection boxes to facilitate the observation of the results when considering scenarios with dense objects. The images unequivocally demonstrated how much more effective the improved model is at object-detecting tasks. The model’s effectiveness combines the advantages of other previously stated models, enabling it to display improved item detection accuracy and efficiency.

### 3.5. Ablation Studies

In this investigation, we carried out ablation experiments to assess and examine the influence of various factors on the functionality of the UAV scene-target-recognition algorithm. The goal of ablation experiments is to thoroughly evaluate the functions of different algorithmic components or parameters under identical experimental circumstances. We performed a detailed study and comparisons of the detection method by gradually eliminating or changing particular components or parameters. We developed a thorough grasp of the algorithm’s operation through the design of these ablation experiments, which also exposing the algorithm’s strengths and weaknesses in a number of areas, offering suggestions for refinement and optimization. In the study, we examined the performance of different models from various perspectives. Table 4 presents a model comparison highlighting the effects of different components on the models, while Figure 11 and Figure 12 illustrate the mAP@0.5 and mAP@0.5:0.95 performances of different models. Furthermore, Figure 13 displays the comparative PR curves of different models.

**Different backbone**: In this study, we conducted modifications to the backbone network of the YOLOv5s model and employed four different lightweight models, namely MobileNetv3, ShuffNetv2s [35], Ghost [36], EfficientNet [37], and SqueezeNet [38]. Under the same input dimensions (640×640), we performed performance evaluations and comparative analyses for each modified model. The experimental findings revealed that the YOLOv5s+MobileNetv3 model exhibited favorable attributes. Despite possessing fewer parameters and computational operations, this model achieved a mAP@0.5 of 0.719 and a processing speed of 222 FPS, demonstrating heightened detection accuracy and accelerated processing speed. In contrast, other selected modified models, including YOLOv5s+ShuffNetv2s, YOLOv5s+Ghost, YOLOv5s+EfficientNet, and YOLOv5s+SqueezeNet, experienced an increase in parameters and computational operations. As a consequence, their corresponding performance levels were somewhat diminished, with mAP@0.5 ranging between 0.682 and 0.704 and mAP@0.5:0.95 ranging between 0.26 and 0.31. Processing speeds varied from 47 FPS to 244 FPS. On the whole, the YOLOv5s+MobileNetv3 model excelled in both detection accuracy and processing speed compared to the other models, all while maintaining efficient resource utilization, characterized by relatively fewer parameters and FLOPs. In summary, the YOLOv5s+MobileNetv3 model outperformed other models in terms of detection accuracy and processing speed, while also demonstrating efficient resource utilization.

**Comparison of Attention Modules**: The coordinate attention (CA), convolutional block attention module (CBAM), and efficient channel attention (ECA) [39] are three distinct attention mechanisms that we designed in order to explore the effects of various attention mechanisms on MobileNetv3. In the MobileNetv3 network, these attentional processes were utilized in place of the traditional SE attentional mechanism, and comparison studies were carried out. According to the experimental findings, adding the CA mechanism to the MobileNetv3 network enhanced the object-identification performance even further. The network may more-accurately record correlations between spatial coordinates by incorporating the CA mechanism, improving the object identification accuracy. In particular, the YOLOv5s+MobileNetv3+CA model demonstrated higher performance with a mAP@0.5 of 0.727 and inference at 166 FPS. In summary, this work carried out comparative studies by integrating several attention mechanisms into MobileNetv3 and proved the efficiency of incorporating the CA mechanism in improving the performance of unmanned aerial vehicle object identification.

**Different IoU algorithms**: We examined the impacts of five alternative IoU algorithms: EIoU [25], CIoU [40], DIoU [41], GIoU [42], and IoU. We only made changes to the IoU algorithm throughout this experiment, keeping the model design and processing resources constant. The experimental findings clearly showed that various IoU algorithms have a substantial impact on the precision and effectiveness of object detection. When measured using the mAP@0.5 metric, the model using the EIoU method had the best accuracy, coming in at 0.752, which was much higher than that of other IoU algorithms. A further demonstration of the EIoU algorithm’s efficacy was the model’s ability to handle 156 FPS while maintaining a constant processing speed of FPS.

**Effectiveness of module**: In response to the particular demands of this work situation, we suggest a number of enhancement methods designed to increase object-detection performance. In order to alleviate the negative effect of deep feature maps on tiny object detection, we first created an ACIFM. The performance of tiny object identification was essentially improved by the ACIFM, which automatically fuses shallow, intermediate, and deep feature maps. These feature maps are then fed into specific small-object-detection heads and a second-level feature map branch. We created an enhanced spatial attention structure (ISPA) to improve the emphasis on small items, further enhancing detection performance, in order to highlight the significance of small objects on the feature map. The YOLOv5s+MobileNetv3+CA+EIoU model was modified utilizing the VariFocal loss (VFL) function to address the problem of sample imbalance in tiny objects. The positive–negative sample imbalance of tiny objects was successfully handled by VFL, significantly improving the detection precision. In the end, we included the aforementioned enhancement techniques into the self-developed SFHG-YOLO final detection network.

Through experimental comparisons and evaluations, the proposed SFHG-YOLO model demonstrated favorable performance with an input image size of 640×640. It achieved an mAP@0.5 score of 0.864 while maintaining an efficient processing rate of 98 FPS. Ablation experiments validated that the integration of this series of improvements resulted in a 7.7% increase in the mAP@0.5 and a 31.3% increase in the mAP@0.5:0.95 compared to the baseline model. Despite the increase in the parameter count and computational complexity due to these enhancements, there was a notable improvement in the accuracy and mAP, effectively striking a balance between model size, precision, and speed. Moreover, the PR curve indicated that the SFHG-YOLO model encompassed other models, further highlighting its performance.

## 4. Conclusions

To address the practical demand for detecting small targets (such as pineapple flowers) in UAV visual estimation of pineapple yield, we designed an SFHG-YOLO algorithm tailored for high-density small target detection. This algorithm leverages MobileNetV3 in combination with the CA module to achieve a lightweight network. Furthermore, we introduced an ACIFM to counter the adverse impact of deep feature maps on small target detection. The ACIFM automatically fuses shallow, intermediate, and deep feature maps, directing attention towards densely populated small target regions, thereby enhancing feature extraction for small targets. Additionally, an ISPA was designed to further emphasize the significance of small targets in the feature map, resulting in enhanced detection performance. To address the issue of imbalanced small target samples, the VFL function was employed to optimize the YOLOv5s+MobileNetv3+CA+EIoU model, strengthening the learning capacity and detection accuracy for small targets. Finally, these improvement measures were integrated into the final detection network, SFHG-YOLO, which we designed. The experimental results demonstrated that the SFHG-YOLO model achieved promising performance with 640×640 image inputs, reaching an mAP@0.5 score of 0.864 while processing 98 FPS. Furthermore, the model maintained low memory requirements, offering a high level of detection accuracy and real-time capabilities. Through verification via ablation experiments, our proposed improvements resulted in a 7.7% increase in the mAP@0.5 and a remarkable 31.3% increase in the mAP@0.5:0.95 compared to the baseline, representing substantial advancements among all models.

Despite the SFHG-YOLO algorithm’s notable successes in the UAV visual-small-object-identification job, there are still many areas that may be further investigated and refined. To further lower the false detection rates and missed detection rates, future work will concentrate on refining the network structure and parameter settings of the algorithm.

## Figures and Tables

**Figure 1 sensors-23-09242-f001:**
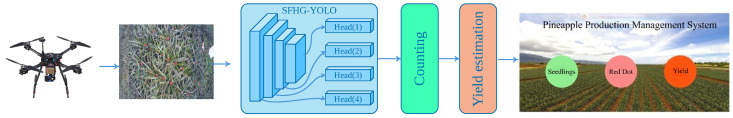
Workflow of pineapple-yield-estimation system.

**Figure 2 sensors-23-09242-f002:**
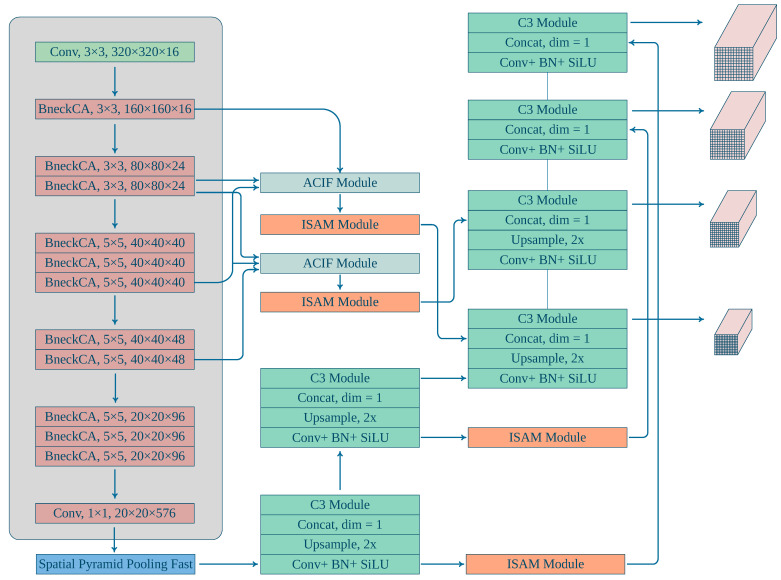
Improved structure of YOLOv5s. We made modifications primarily to the backbone and detection head of the model, introducing the Bneck module from MobileNetv3, and incorporating the coordinate attention (CA) mechanism into the Bneck module.

**Figure 3 sensors-23-09242-f003:**
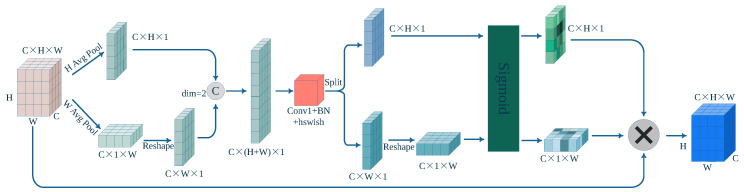
Structure of the CA module. The main idea is to embed the location information into the channel’s attention. It enables the lightweight network to operate with attention over a larger area.

**Figure 4 sensors-23-09242-f004:**
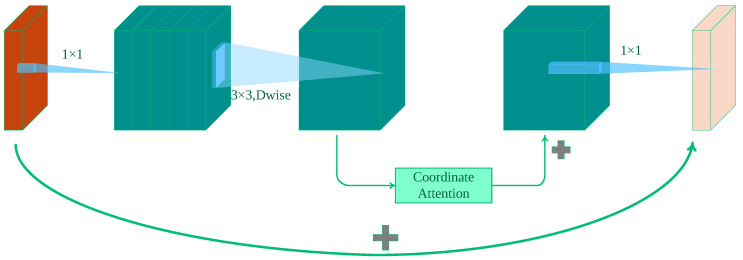
The enhanced Bneck module (BneckCA) of MobileNetV3. We modified the structure highlighted in green shadow in the diagram, which originally was an SE module, and replaced it with a CA module (BneckCA).

**Figure 5 sensors-23-09242-f005:**
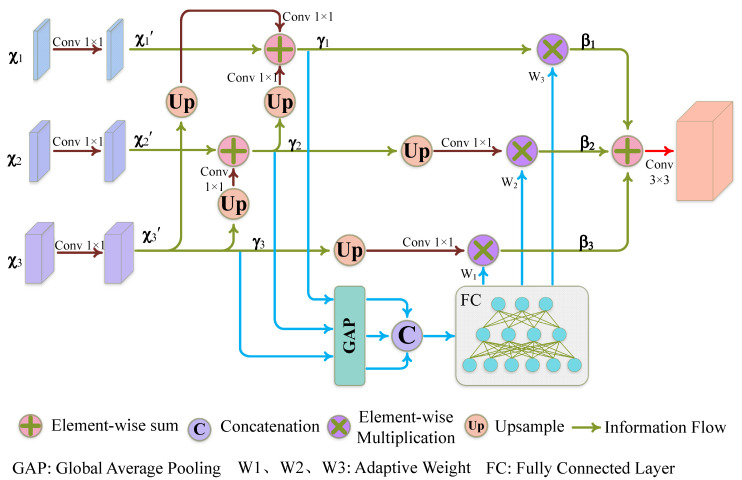
Adaptive context information fusion module. We selected information from various levels and determined the weight coefficients for each level using fully connected layers. Ultimately, we integrated the information obtained from different scales.

**Figure 6 sensors-23-09242-f006:**
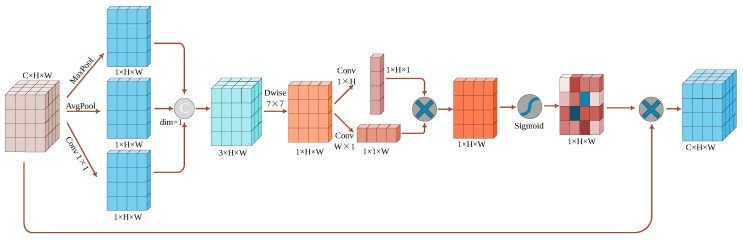
Improved spatial attention module (ISPA).

**Figure 7 sensors-23-09242-f007:**
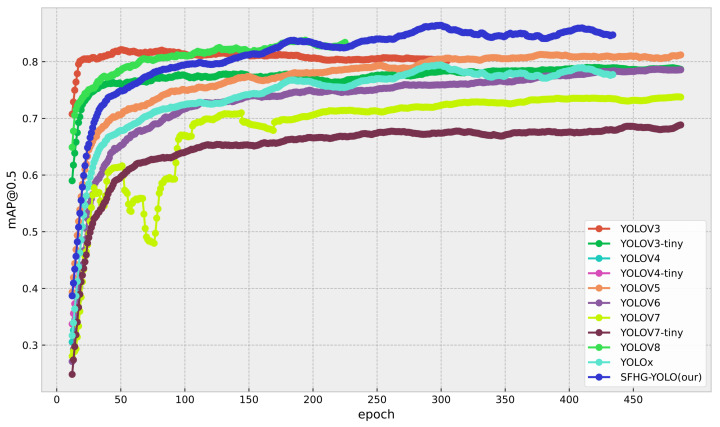
mAP@0.5 for the comparison experiment.

**Figure 8 sensors-23-09242-f008:**
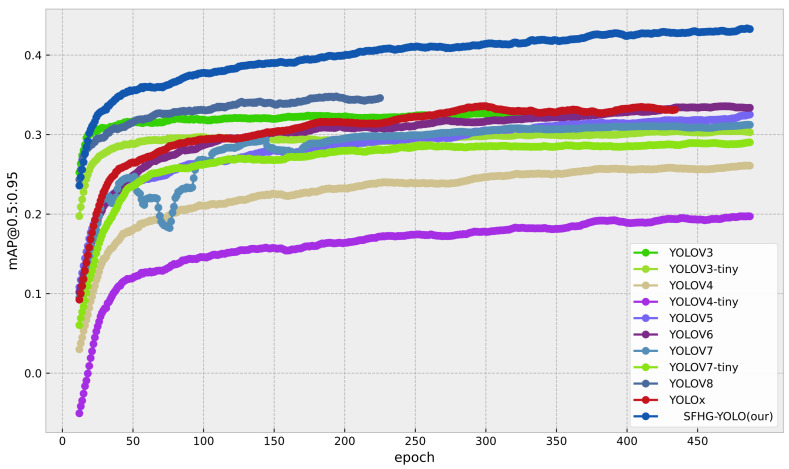
mAP@0.5:0.95 for the comparison experiment.

**Figure 9 sensors-23-09242-f009:**
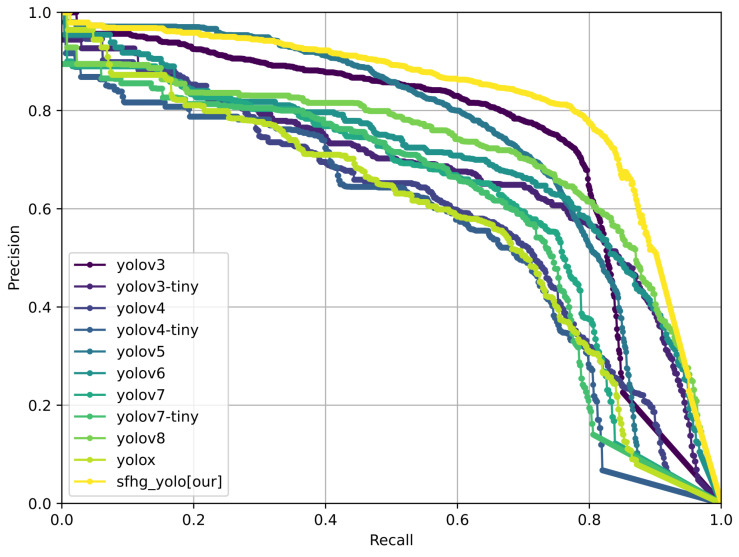
Precision–recall curve for the comparison experiment.

**Figure 10 sensors-23-09242-f010:**
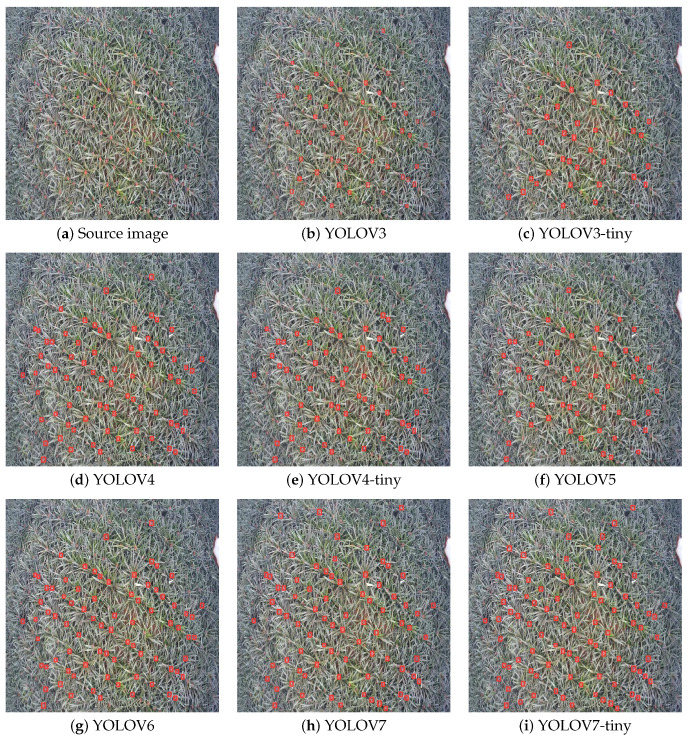
Visualization comparison of different models.

**Figure 11 sensors-23-09242-f011:**
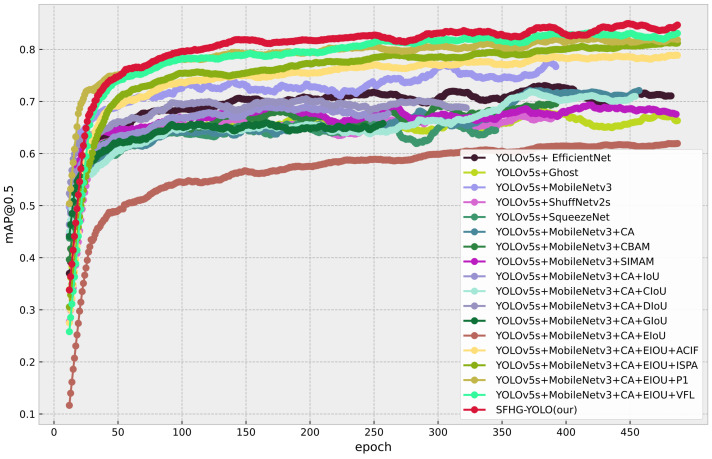
mAP@0.5 for the ablation experiment.

**Figure 12 sensors-23-09242-f012:**
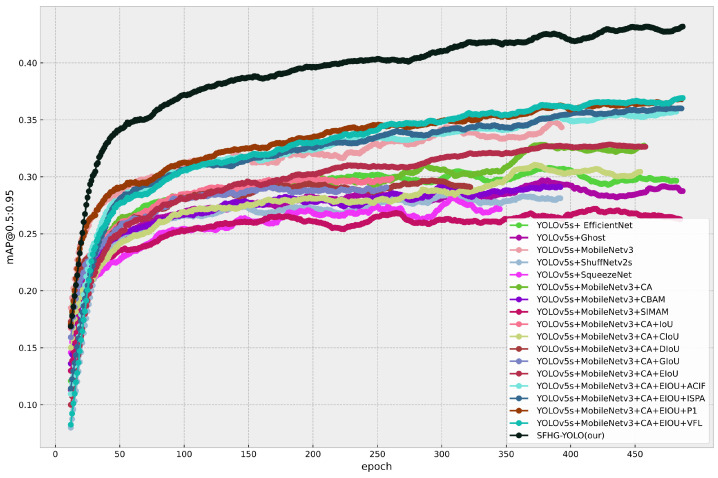
mAP@0.5:0.95 for the ablation experiment.

**Figure 13 sensors-23-09242-f013:**
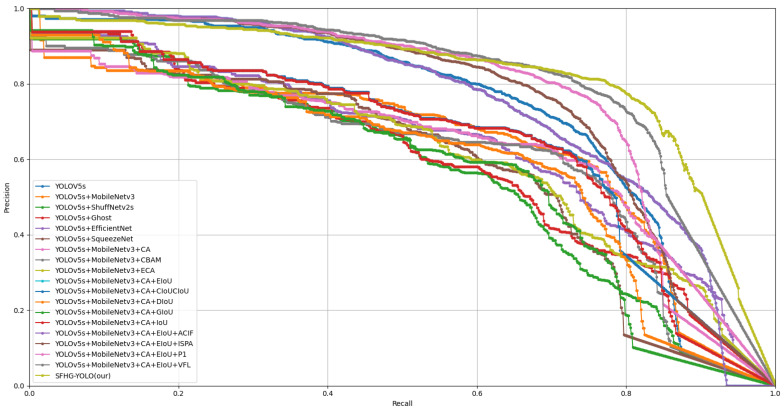
Precision–recall curve for the ablation experiment.

**Table 1 sensors-23-09242-t001:** Experimental environment and related configurations.

Configuration	Amount	Value
CPU	2	Intel(R) Xeon(R) Gold 6248R @ 3.0 GHz
GPU	2	Nvidia GeForce RTX 3090 GPU
CUDA	-	CUDA Toolkit 11.7
Cudnn	-	Cudnn 8.8.0
Memory	2	128 GB RAM
Mechanical hard disk	7	1 TB
Operating system	-	Windows Server 2019 Standard
Deep learning framework	-	Pytorch1.13.0

**Table 2 sensors-23-09242-t002:** Training params’ setting.

Configuration	Parameters
Epochs	500
Batch size	16
Learning rate	0.001
SGD	True
ConsineAnnealing	True
Weight decay	0.05
Input size	640 × 640
Workers	12

**Table 3 sensors-23-09242-t003:** Performance comparison of different models.

No.	Size	Param (M)	FLOPs (G)	mAP@0.5	mAP@0.5:0.95	FPSs	MMRs (M)
YOLOV3	640 × 640	123.5	155.3	0.804	0.32	25	231.1
YOLOV3-tiny	640 × 640	17.4	13.5	0.781	0.30	238	33.7
YOLOV4	640 × 640	70.3	63.9	0.50	0.25	16	247.0
YOLOV4-tiny	640 × 640	8.08	5.87	0.43	0.19	130	18.6
YOLOV5	640 × 640	8.7	15.9	0.802	0.32	166	27.6
YOLOV6	640 × 640	19.4	47.1	0.785	0.30	54	29.7
YOLOV7	640 × 640	106.5	38.6	0.732	0.31	66	147.3
YOLOV7-tiny	640 × 640	14.3	8.7	0.698	0.29	212	23.6
YOLOV8	640 × 640	12.1	9.1	0.806	0.34	185	52.6
YOLOX	640 × 640	26.8	9.0	0.792	0.33	85	42.1
SHFP-YOLO (our)	640 × 640	2.69	6.34	0.864	0.42	98	9.7

**Table 4 sensors-23-09242-t004:** Results of ablation experiment.

No.	Size	Param	FLOPs	mAP@0.5	mAP@0.5:0.95	FPSs	MMRs (M)
YOLOv5s	640 × 640	8.70	15.9	0.802	0.32	166	27.6
YOLOv5s+MobileNetv3	640 × 640	1.73	3.2	0.719	0.31	222	7.2
YOLOv5s+ShuffNetv2s	640 × 640	4.13	9.7	0.682	0.28	133	16.4
YOLOv5s+Ghost	640 × 640	3.39	5.5	0.683	0.28	123	13.6
YOLOv5s+EfficientNet	640 × 640	20.74	52.0	0.704	0.29	47	6.5
YOLOv5s+SqueezeNet	640 × 640	1.25	2.5	0.695	0.26	244	6.4
YOLOv5s+MobileNetv3+CA	640 × 640	1.36	2.9	0.727	0.32	166	5.8
YOLOv5s+MobileNetv3+CBAM	640 × 640	1.26	2.8	0.711	0.29	196	5.4
YOLOv5s+MobileNetv3+ECA	640 × 640	1.26	2.8	0.683	0.26	178	5.4
YOLOv5s+MobileNetv3+CA+EIoU	640 × 640	1.36	2.9	0.752	0.32	156	5.8
YOLOv5s+MobileNetv3+CA+CIoU	640 × 640	1.36	2.9	0.712	0.31	156	5.8
YOLOv5s+MobileNetv3+CA+DIoU	640 × 640	1.36	2.9	0.699	0.29	156	5.8
YOLOv5s+MobileNetv3+CA+GIoU	640 × 640	1.36	2.9	0.684	0.29	156	5.8
YOLOv5s+MobileNetv3+CA+IoU	640 × 640	1.36	2.9	0.701	0.30	156	5.8
YOLOv5s+MobileNetv3+CA+EIoU+ACIF	640 × 640	3.10	6.3	0.791	0.35	119	9.4
YOLOv5s+MobileNetv3+CA+EIoU+ISPA	640 × 640	2.65	3.0	0.815	0.35	128	8.6
YOLOv5s+MobileNetv3+CA+EIoU+P1	640 × 640	2.68	6.1	0.834	0.38	121	8.8
YOLOv5s+MobileNetv3+CA+EIoU+VFL	640 × 640	1.36	2.9	0.851	0.37	156	5.8
SFHG-YOLO(our)	640 × 640	2.69	6.3	0.864	0.42	98	9.7

## Data Availability

The data presented in this study can be requested from the corresponding author, and these data are not currently available for public access.

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
