# Peer review of "SFHG-YOLO: A Simple Real-Time Small-Object-Detection Method for Estimating Pineapple Yield from Unmanned Aerial Vehicles"

_sensors, 2023, doi:10.3390/s23229242_

Round 1

Reviewer 1 Report

The paper uses Yolov5 as a baseline model for improvement and applies it to tiny objects for detection, through the comparative experimental results, the author's proposed model is improved compared with the latest Yolov7, yolov8, which is a very good work.

The authors' contributions, and whether there is a conflict of interest are to be stated later. Rather than after the author's affiliation.

I think this article is ready to be accepted after the above comment has been modified.

Minor editing of English language required

Author Response

Dear Reviewer,

Thank you for your valuable feedback and suggestions. We appreciate your thorough review, insightful comments, and constructive recommendations, which have been instrumental in enhancing the quality of our manuscript.

We have made the necessary revisions based on your suggestions, The improved sections are highlighted with yellow markers. the revised manuscript has been uploaded as an attachment for your reference, and we are grateful for your review.

Sincerely, [Wang Tao]

Reviewer 2 Report

The paper proposes changes to YOLOv5s to make it more effective for detecting small items in the frame captured by a UAV. The authors use the pineapple yield estimation scenario to highlight the proposed method's power. The overall structure of the paper is good. There are a few suggestions and comments that may help enhance the quality of the article:

- It would be helpful to understand why high-frequency detection is required in the given problem. It is mentioned that "the drone flies according to commands from the base station while simultaneously recording films and taking pictures during the flying missions," so the UAV does not make decisions based on its detection results and does not need immediate yield estimation (or yield estimation on every frame), so I would like to understand why in such a problem heavier but more powerful methods such as two-stage detection methods (e.g., Mask R-CNN with a transformer backbone) would not work. Many powerful architectures can still process a few frames per second and are much more accurate than a single-stage method can achieve with any enhancements.

- The paper is generally written well; however, I suggest another round of proofreading by a native English speaker and a detailed check by the authors for typographical issues. It will help catch problems such as referencing the incorrect figure (e.g., on line 250). At the minimum, a check by software (e.g., Grammarly) can catch some of the typos, sudden breaks in the sentences (e.g., paragraph of 2.2.2), and more minor issues (e.g., not putting a space after punctuation marks). I would also suggest removing words and phrases such as "admirably," "exceptional," etc., for your own work. The flow can also definitely improve. For example, continuing the text about Equation 13 after Equations 10-12, or putting Figures closer to where they are mentioned, would allow the user to read continuously without jumping back and forth between paragraphs and pages. Finally, every symbol needs to be explained to the reader. A lot of symbols in equations are missing a detailed explanation.

- The information on the dataset should be expanded. How large is the dataset? The number of frames, minutes, etc.? Was the annotation manual? How was the training/validation/testing division done?

- It seems that the paper has trouble with metrics. The definitions of accuracy and recall on lines 396-400 are incorrect. Additionally, on line 402, it says, "By computing the area under the precision-recall curve, Average Precision (AP) evaluates the precision of a model," which seems that the area under the precision-recall curve (AUC) is confused (or considered connected) with AP. Technically, AUC is not even computed in the results here, so not sure why it is being mentioned at all. On line 408, it says, "The terms accuracy and completeness (P and R, respectively) correspond to the model's precision and recall," but I do not see the word "completeness" or "R" anywhere in the paper, so I assume it was a bad copy/paste from somewhere else.

- There is some discussion on defining recall, but I did not see any recall rates reported. If no recall is computed, its definition can also be omitted from the text. However, overall, I would like to see the recall rates and the ROC AUC curve for the methods. They can provide helpful additional context on top of the mAP.

- In general, many equations are poorly described, and their symbols are not explained. It also seems some equations have typos and errors, which is hard to tell due to the lack of explanations.

The paper is generally written well; however, I would suggest another round of proofreading by a native English speaker and a detailed check by the authors for typographical issues. At the minimum, a review by software (e.g., Grammarly) can catch some of the typos, sudden breaks in the sentences (e.g., paragraph of 2.2.2), and more minor issues (e.g., not putting a space after punctuation marks). I would also suggest removing words and phrases such as "admirably", "exceptional", etc., for own work.

Author Response

Dear Reviewer,

Thank you very much for your valuable feedback. Your insights and suggestions have been crucial in refining and enhancing our paper. We have carefully considered your comments and made the necessary revisions accordingly, with the hope of receiving your approval. The revised sections are highlighted in yellow within the updated version. I will address each of your points individually to provide a clear response to your concerns:

1. Regarding your query about why we opted not to use a two-stage object detection algorithm, I sincerely apologize for the lack of clarity in our previous explanation. In response, we have made clarifications in the second paragraph of the Introduction to accurately portray the practical requirements of the problem. As per the project's specifications, real-time calculations and yield estimation are needed during pineapple yield assessment flights, along with immediate data transmission to meet client demands. We genuinely appreciate your meticulous review, timely identification of the issue, and the guidance you've provided. We regret any previous inadequacies in our description and are genuinely thankful for your insights and suggestions.

2. Concerning the grammar and fluency of the manuscript, we have meticulously proofread the paper as per your suggestions. We have also removed words such as "admirably" and "exceptional" from the text. Additionally, we have repositioned the images of Equations 10-13 to enhance the coherence and readability of the paper. While we have taken steps to improve, we acknowledge that there might still be areas for enhancement. Thus, we eagerly await any further guidance you can provide to help us elevate the quality of the manuscript.

3. In response to your inquiry about the dataset, we have provided an expanded description in Section 3.1 of the Experimental Setup. Furthermore, we wish to confirm that the data used in the experiments was manually annotated.

We have addressed grammatical issues and other concerns to the best of our ability, but we understand there may still be room for improvement. As a result, we are eagerly looking forward to any additional suggestions you can offer to assist us in further refining the manuscript.

Once again, we deeply appreciate your guidance and invaluable recommendations.

Warm regards,
[Wang Tao]

Reviewer 3 Report

1. To reduce the computational complexity and memory requirements, some common techniques have been proposed, such as weight quantization, network pruning, transferred learning. If you feel appropriate, you may add some of these references to review and discuss them. It is highly desirable to apply these techniques to simplify your proposed SFHG-YOLO model.

[1] Tang Z, Luo L, Xie B, et al. Automatic sparse connectivity learning for neural networks[J]. IEEE Transactions on Neural Networks and Learning Systems, 2022.

[2] Huang Q. Weight-quantized squeezenet for resource-constrained robot vacuums for indoor obstacle classification[J]. AI, 2022, 3(1): 180-193.

[3] Kundu A, Mellempudi N K, Vooturi D T, et al. AUTOSPARSE: Towards Automated Sparse Training of Deep Neural Networks[J]. arXiv preprint arXiv:2304.06941, 2023.

[4] Hu W, Che Z, Liu N, et al. : Channel Pruning via Class-Aware Trace Ratio Optimization[J]. IEEE Transactions on Neural Networks and Learning Systems, 2023.

[5] Zheng J, Lu C, Hao C, et al. Improving the generalization ability of deep neural networks for cross-domain visual recognition[J]. IEEE Transactions on Cognitive and Developmental Systems, 2020, 13(3): 607-620.

[6] Hao C, Chen D. Software/hardware co-design for multi-modal multi-task learning in autonomous systems[C]//2021 IEEE 3rd International Conference on Artificial Intelligence Circuits and Systems (AICAS). IEEE, 2021: 1-5.

[7] D. N. Endrawati, S. I. Ibad, I. Syafalni, N. Sutisna, R. Mulyawan and T. Adiono, "YOLOv3- Tiny's Weight Size Reduction using Pruning and Quantization," 2021 15th International Conference on Telecommunication Systems, Services, and Applications (TSSA), Bali, Indonesia, 2021, pp. 1-5, doi: 10.1109/TSSA52866.2021.9768258.

2. SqueezeNet is a deep learning architecture specifically designed for efficient convolutional neural network (CNN) model inference on resource-constrained devices, such as mobile phones and embedded systems. It was introduced as a solution to address the challenges of deploying deep neural networks on devices with limited computational capabilities and memory. I expect the authors to add YOLO+SqueezeNet into Table 4.

3. I expect the authors to add one column in Table 4 to add the memory requirement for each model in this table.

acceptable

Author Response

Dear Reviewer,

We extend our heartfelt gratitude for your invaluable suggestions. Your insights on common techniques to reduce memory requirements are highly appreciated, and they will indeed guide our future endeavors. Additionally, we are profoundly thankful for your advice, which has significantly contributed to the refinement of our manuscript.

Following your guidance, we have incorporated a new column (MMR) in both Table 3 and Table 4 to provide additional information about the memory requirements of different models in Float32 precision. Furthermore, we have added a new row for YOLOv5s+SqueezeNet in Table 4 to facilitate a comparison of the effects of different backbone networks. We have also made further improvements to the grammar and introduction of the paper. The revised manuscript, with highlighted improvements in yellow, has been attached for your reference. We eagerly anticipate your continued invaluable suggestions, which will undoubtedly aid us in further enhancing our manuscript. Once again, we express our heartfelt appreciation for your attentive guidance.

Thank you sincerely for your support and guidance.

Best regards,
[Wang Tao]

Round 2

Reviewer 2 Report

The revised version has noticeably improved and the authors have addressed most of the comments and concerns. The writing is better, the metric definitions are correct and there is more explanation of obscure parts now. There are only a few minor suggestions that can enhance the quality of the publication further and help the future readers:

- Some abbreviations are used in the text without using the full name (e.g., FPN). It would be beneficial to a reader who is new to this field to have the full name on the first use.

- There are still some minor typographical issues (for example, not putting space before citation brackets, capitalized words in the middle of sentence (e.g., line 412), etc.). I suggest using any spell-checking software, they can help catch those issues.

Author Response

Dear Reviewer,

Thank you for your valuable feedback once again. We greatly appreciate your thorough review of our revised manuscript and the valuable suggestions you provided.

We have carefully made revisions to the manuscript to further improve its quality. The revised sections have been highlighted with yellow markers, and they are included in the supplementary materials.

1. Firstly, we have carefully reviewed the abbreviations in the manuscript, including "FPN," to ensure that the full names are spelled out upon first mention. This will help readers who may not be familiar with this field.

2. Secondly, we have conducted final proofreading of the manuscript to address any remaining formatting issues, such as spacing before citation brackets and capitalization in the middle of sentences. We have used spell-checking software to identify and correct these issues.

Your feedback has played a crucial role in improving the quality and readability of our manuscript, and we sincerely appreciate your thorough review of our work.

Thank you once again for your time and expertise in reviewing our paper. We believe that our revisions have strengthened the manuscript, and we hope it now meets the high standards of the journal.

Best regards,
[Wang Tao]

Reviewer 3 Report

acceptable

Author Response

Dear Reviewer,

Thank you for once again carefully reviewing our manuscript and providing valuable feedback. We greatly appreciate your expert insights, as your suggestions are crucial for improving the quality of our paper.

Based on your recommendations, we have made the necessary revisions to the paper and have marked the improved sections in the revised version. Attached is the modified manuscript for your reference.

We will ensure that we provide full names when using abbreviations for the first time in the text to enhance reader comprehension. We will also further review spelling and grammar issues in the manuscript to ensure its language quality.

Thank you once again for taking the time to review our paper; your feedback is of utmost importance to our research efforts. If you have any further suggestions or questions regarding the revised manuscript or if you would like to discuss any issues in more detail, we would greatly appreciate your expert guidance and look forward to your response.

Sincerely,
[Wang Tao]
